# Future Pharmacotherapy for Sensorineural Hearing Loss by Protection and Regeneration of Auditory Hair Cells

**DOI:** 10.3390/pharmaceutics15030777

**Published:** 2023-02-26

**Authors:** Mami Matsunaga, Takayuki Nakagawa

**Affiliations:** Department of Otolaryngology, Head and Neck Surgery, Graduate School of Medicine, Kyoto University, Kyoto 606-8501, Japan

**Keywords:** clinical trial, cochlea, hair cell, hearing loss, regeneration, protection

## Abstract

Sensorineural hearing loss has been a global burden of diseases for decades. However, according to recent progress in experimental studies on hair cell regeneration and protection, clinical trials of pharmacotherapy for sensorineural hearing loss have rapidly progressed. In this review, we focus on recent clinical trials for hair cell protection and regeneration and outline mechanisms based on associated experimental studies. Outcomes of recent clinical trials provided valuable data regarding the safety and tolerability of intra-cochlear and intra-tympanic applications as drug delivery methods. Recent findings in molecular mechanisms of hair cell regeneration suggested the realization of regenerative medicine for sensorineural hearing loss in the near future.

## 1. Sensorineural Hearing Loss: Serious Healthcare Problem

Hearing loss has been a global disease burden for decades [1]. Approximately 466 million people have a hearing disability. Hearing loss reduces functioning in everyday tasks [2] and is a causative factor of clinically significant depression, anxiety, and stress symptoms [3]. In addition, a world-shaking study identified a close relationship between hearing loss and dementia [4], increasing the demand for the development of novel therapeutics for hearing loss worldwide. The most common form of hearing loss is sensorineural hearing loss (SNHL), which is caused by a range of genetic and environmental factors, including noise exposure that damage the inner ear or auditory nerve. A natural aging process also causes SNHL [5]. Genome-wide association studies for age-related SNHL demonstrated several candidate genes that accelerate SNHL due to aging [6]. The available pharmacotherapy for SNHL is currently limited in clinical settings. Systemic corticosteroids have long been an option for the treatment of acute SNHL, but their efficacy is limited [7,8]. As an alternative to systemic corticosteroids, intratympanic corticosteroid treatment by direct injection into the middle ear has recently gained popularity for the treatment of acute SNHL, including sudden deafness [9,10]. Some reports have indicated that the efficacy of topical application is superior to that of systemic application, but the efficiency rate is not satisfactory.

Currently, no pharmacotherapeutic options are available for stable SNHL. Stable SNHL is the condition of SNHL showing no changes in hearing levels at any frequency by standard audiometric measures for six months, which includes acute SNHL previously treated with no response to corticosteroids [11]. The main treatment of stable SNHL is medical devices, such as hearing aids and cochlear implants. These hearing devices are beneficial, but they have significant drawbacks. Hearing aids are poorly effective for sound perception in the presence of background noise [12], and cochlear implants require surgery and can provide rudimentary sound perception. A recent publication of healthcare economic modeling reported that the replacement of hearing aids and cochlear implants in regenerative medicine has a huge impact on economics [13]. If this replacement is achieved, regenerative medicine for SNHL will yield a benefit of £15,000 per treatment even with a 50% efficacy rate [13]. Consequently, the need to develop novel pharmacotherapies for SNHL is increasing.

## 2. Therapeutic Targets for Sensorineural Hearing Loss

Sound stimuli are converted into neural signals in the cochlea of the inner ear. The vibration of otic ossicles, which transmit sound vibration from the tympanic membrane to the cochlea, generates sound waves in the cochlea. Sound waves cause a vibration of the cochlear sensory epithelium, in which sensory hair cells (HCs) form four rows along the cochlear coil. Cochlear sensory hair cells are divided into two types: inner hair cells (IHCs) and outer hair cells (OHCs). The IHCs are distributed in a single row and play a central role in the transmission of sound stimuli to the auditory primary neurons, namely the spiral ganglion neurons (SGNs), which are located in the modiolus of the cochlea. The OHCs are distributed in three rows on the outside of the IHC row and amplify cochlear sensory epithelium vibration [14,15,16]. The sound vibration causes distortion of the hair bundles on the apical surface of the IHCs, resulting in their depolarization. Subsequently, neurotransmitters are released from the presynaptic ribbons at the bottom of the IHCs to the postsynaptic patches in the afferent dendrites of SGNs. This synaptic contact is called the ribbon synapse, which is characteristic of the sensory system, retina, and inner ear [17] and is the most vulnerable site in the cochlea for noise trauma and aging [18]. In the cochlear lateral wall, the stria vascularis and spiral ligament are present and play a crucial role in the generation of endocochlear potential, which is necessary for the depolarization of IHCs. The stria vascularis also plays a key role in regulating the transport of molecules from the bloodstream into the cochlea, also known as the blood–labyrinthine barrier. Endocochlear potential is also required for OHC electromotility [19], which is necessary for modulating cochlear sensory epithelium vibration.

Damage to all cellular components in the cochlea can cause SNHL. The loss of HCs has long been considered a major cause of SNHL [20]. Recently, the ribbon synapse between IHCs and SGNs has gained considerable attention as a therapeutic target for SNHL [21,22]. The progress in causative genes for SNHL has revealed many deafness genes [https://hereditaryhearingloss.org/ (14 February 2023)]. Several deafness genes are associated with hair bundle development and functioning and synaptic transmission [6,17]. Hereditary hearing loss is a major cause of SNHL, and recent progress in gene therapy is opening therapeutic opportunities in patients with hereditary hearing loss [23]. In addition, several candidate genes that accelerate age-related, noise-induced, and drug-induced SNHL have been demonstrated [6]. Therefore, gene therapy can be a therapeutic option for the prevention of SNHL due to various causes. In addition, accompanied by the popularization of genetic screening, gene therapy can contribute to the development of individualized therapy for SNHL. Regardless of the etiology, HCs are the major targets for the treatment of SNHL.

## 3. Recent Clinical Trials for the Prevention of Sensorineural Hearing Loss

Recently, several clinical trials for SNHL have been conducted [11,24]. Recent clinical trials for SNHL can be divided into two categories: preventing SNHL and hearing regeneration. Preventing SNHL may be more realistic than regeneration, while the therapeutic time window for regeneration is considerably wider than that for prevention.

HC protection has long been investigated for decades using a variety of experimental models [25,26,27]. Ototoxic chemicals, aminoglycosides, and ototoxic antibiotics have widely been used for inducing HC damage in both in vivo and in vitro experiments. Platinum compounds, such as cisplatin and carboplatin, which are effective against a wide range of malignancies, have been used to cause HC degeneration. Noise trauma has frequently been used in in vivo experiments. These animal models for SNHL have played important roles in developing novel therapeutics for SNHL. Although detailed mechanisms of these causes for HC damage differ, the generation of reactive oxygen species (ROS) is a common mechanism. ROS-induced cellular stress leads to HC death. To promote HC survival, molecular pathways for ROS generation and consecutive HC death have extensively been investigated. Based on these previous experimental studies, clinical trials aimed at preventing SNHL have been conducted.

Le Prell (2021) performed a comprehensive search of clinical trials for noise-induced hearing loss, drug-induced hearing loss, stable SNHL or age-related hearing loss, and sudden SNHL in the National Library of Medicine and identified 61 clinical trials [11]. The most active clinical trial program appears to be that of drug-induced hearing loss [11]. Among the 30 clinical trials for drug-induced hearing loss, cisplatin-induced hearing loss was the subjective disease in 26. We, thus, focused on recent clinical trials for cisplatin-induced hearing loss.

In Table 1, the status of recent clinical trials for cisplatin-induced hearing loss is summarized. Updated information on each clinical trial is available from the website [https://clinicaltrials.gov/ct2/home (14 February 2023)]. Cisplatin is widely used for the treatment of malignant neoplasms as well as pediatric cancers. Children are at greater risk of developing hearing loss than adults are, with dire consequences for speech development and social integration. Among patients with SNHL, the proportion of cisplatin-induced hearing loss is comparatively small, but over 50% of patients who undergo cisplatin therapy acquire SNHL [28,29]. Considering the years with disability, the impact of the prevention of cisplatin-induced hearing loss is not negligible. In cisplatin-induced hearing loss, the main target is HCs, because HCs are one of the most susceptible cells to cisplatin toxicity in the cochlea [30]. Therefore, agents that have protective effects against cisplatin toxicity and do not diminish cisplatin efficacy in malignant tumors are desired. In September 2022, sodium thiosulfate (Table 1) received its first approval in the USA for reducing the risk of ototoxicity associated with cisplatin in pediatric patients 1 month of age and older with localized, non-metastatic solid tumors [31]. Results of two Phase 3 trials were published. In the SIOPEL 6 trial (NCT00652132), hearing loss occurred in 18 of 55 children (33%) in the cisplatin–sodium thiosulfate group, compared with 29 of 46 (63%) in the cisplatin-alone group [32]. In the COG ACCL0431 trial (NCT00716976), hearing loss occurred in 14 of 49 children (29%) in the cisplatin–sodium thiosulfate group, compared with 31 of 55 (56%) in the cisplatin-alone group [33].

Clinical trials for a statin (atorvastatin), calcineurin antagonist (SENS-401), and glutathione mimic (SPI-1005) are ongoing (Table 1). These agents have also exhibited potential as therapeutics for SNHL due to noise and/or aging [34,35,36], which suggests the possible expansion of their clinical application for SNHL due to various etiologies. Furthermore, the mechanism for the efficacy of statins and glutathione mimics, the reduction of cellular stress, is very common for HC protection against various types of injuries. In addition, these agents exhibit efficacy after systemic application and oral application has been used in clinical trials. If no serious side effects are identified in clinical trials, these agents could be used as baseline therapeutics for SNHL in future.

Among these agents, clinical trials of atorvastatin have progressed the most [11,24]. Statins are HMG-CoA inhibitors widely used in patients with hyperlipidemia. Several statins are FDA-approved. Therefore, if the dose and application route are similar to those used in hyperlipidemia, clinical trials can be planned for off-label use, meaning that several preclinical studies can be skipped. The lack of appropriate animal models has been an obstacle in the development of therapeutics for cisplatin-induced hearing loss. However, a recent study succeeded in establishing an excellent animal model that reflects the clinical setting [30]. In addition, this animal model has been validated in studies of cisplatin-induced hearing loss in animals [37] and humans [38]. This animal model may serve as a standard model for investigating cisplatin-induced hearing loss.

## 4. Recent Clinical Trials for Regenerative Medicines

In comparison with the prevention of SNHL, clinical trials of regenerative medicine for SNHL are more challenging. Recent clinical trials aimed at hearing regeneration have focused on two targets: HCs and ribbon synapses (Table 2) [24]. Most SNHL cases result from irreversible damage to the HCs. Since the mature mammalian cochlea has virtually no capability for HC regeneration, SNHL due to HC loss is permanent [39,40]. However, several decades have demonstrated the potential for HC regeneration in mammalian cochleae [6]. Although the development of pharmacotherapy that induces HC regeneration seems moonshot, its impact will be incredible [13]. Regeneration or repair of HC structures, including ribbon synapses, is a more practical target [41]; however, its impact on hearing benefits in vivo is still unclear. In clinical trials aimed at HC regeneration (Table 2), the fate conversion of supporting cells (SCs) to HCs is the predominant mechanism for HC regeneration [24]. For this purpose, the forced expression of ATOH1, which is a basic helix–loop–helix transcription factor that plays a critical role in the development and regeneration of HCs [42], by gene transfer using an adenovirus vector or pharmacological inhibition of Notch signaling, is utilized (Table 2).

### 4.1. Clinical Trial of ATOH1 Gene Therapy

The human adenoviral vector serotype 5 was used for ATOH1 gene transfer to SCs in a clinical trial (NCT02132130). A results summary is available on the website [https://www.novctrd.com/ctrdweb/patientsummary/patientsummaries?patientSummaryId=680 (14 February 2023)]. Intracochlear application was used as the drug application route. No serious adverse events were found in 22 participants, but slight hearing loss was identified in 32% (7/22). In hearing assessments using pure tone audiometry, no meaningful increase in hearing was identified. Based on these results, the trial was suspended but not canceled. The most important finding of this trial is that the intracochlear delivery of adenovirus vectors is tolerable, which may aid the progression of future clinical trials using either intracochlear application or adenovirus vectors.

The scientific background of this clinical trial probably included two experimental studies: Izumikawa et al. (2005) [43] and Kraft et al. (2011) [44]. Izumikawa et al. (2005) demonstrated HC replacement and hearing recovery following ATOH1 gene transfer using an adenovirus vector in mature guinea pigs [43]. Kraft et al. (2011) used adult mice as experimental animals and adenovirus vectors were administered from the posterior semicircular canal [44]. Significant hearing recovery was identified at low frequencies in ears treated with adenovirus vectors [44]. Controversially, Atkinson et al. (2014) reported no significant hearing recovery in mature guinea pigs after ATOH1 gene transfer using adenovirus vectors, although HC marker-positive cells showed a significant increase [45]. Although functional recovery varied among the three studies, HC marker-positive cells were generated in all studies, even when adult mammals were used. We should pay attention to the fact that ATOH1 gene transfer was performed shortly after HC loss in these studies. Considering the clinical setting, interventions immediately after HC loss are not realistic. The degree of damage to the cochlear sensory epithelium and consecutive damage to SGNs are also important. Izumikawa et al. (2008) demonstrated that ATOH1 gene transfer resulted in no HC regeneration in the severely damaged sensory epithelium of adult guinea pigs [46], which suggests that whether responsible SCs remain in the cochleae of patients may affect outcomes in clinical settings. In the future, the eligibility criteria for clinical trials will become important for achieving satisfactory outcomes in functional assessments. In addition, persistent expression of ATOH1 in HCs was reported to be a cause of HC death in mouse cochleae [47]; therefore, how to shut ATOH1 down might be included in future challenges.

### 4.2. Clinical Trials of Notch Signaling Inhibition

Two-phase 1/2 clinical trials of γ-secretase inhibitors aimed at inhibiting Notch signaling were completed (Table 2, NCT04462198; EudraCT number 2016-004544-10). The results of a clinical trial for LY-3056480 (EudraCT number 2016-004544-10) are available [https://www.clinicaltrialsregister.eu/ctr-search/trial/2016-004544-10/results (14 February 2023)], which show that trans-tympanic injection (three administrations, one week apart) at the highest dose of 250 micrograms of LY-3056480 was safe and well tolerated. A phase 2 double-blind, randomized, placebo-controlled trial for this compound was registered as NCT05061758; however, the present status for recruitment is “withdrawn.” The outcomes of hearing recovery in these clinical trials have not yet been reported.

The efficacy of the pharmacological inhibition of Notch signaling by γ-secretase inhibitors was first reported in neonatal mice [48,49]. The capability of HC regeneration by pharmacological inhibition of Notch signaling was demonstrated in adult guinea pigs, although the number of newly generated HC marker-positive cells was limited [50]. For the ATOH1 forced expression by γ-secretase inhibitors, activation of Notch signaling in the cochlear sensory epithelium is necessary. Hori et al. (2007) showed the activation of Notch signaling in SCs by immunohistochemistry two days after ototoxic treatment [50] in which the expression levels of Notch1 and Jagged 1 in SCs seven days after treatment returned to the control level [51]. Mizutari et al. (2013) demonstrated the activation of Notch signaling using quantitative real-time polymerase chain reaction immediately after noise exposure [51]. The expression level of Hes5 was downregulated to the control level three days after noise exposure [51]. Hence, the therapeutic time window for pharmacological inhibition of Notch signaling by γ-secretase inhibitors is limited to a few days after HC damage. From this perspective, γ-secretase inhibitors alone may not be effective in patients with stable SNHL. The duration after the onset of SNHL could be a key element in the recruitment of patients for this treatment.

Functional recovery by γ-secretase inhibitors has been observed in experimental studies on adult mammals [51,52]. Mizutari et al. (2013) demonstrated that the local application of the γ-secretase inhibitor LY411575 induced HC regeneration and hearing recovery in a mouse model of noise-induced hearing loss [51]. In this study, OHC regeneration and approximately 10 dB recovery in auditory brainstem response (ABR) thresholds were observed [51]. Distortion-product optoacoustic emissions (DPOAEs), which reflect OHC functions, were not evaluated by Mizutari et al. (2013). Tona et al. (2014) investigated functional and histological restoration using the γ-secretase inhibitor MDL28170 in an adult guinea pig model of noise-induced hearing loss [52]. Histologically, a significant increase in OHCs was demonstrated [52] similar to Mizutari et al. (2013). In functional assessments, significant recovery was identified in ABR threshold shifts but not in DPOAEs [52]. These results illustrate the discrepancy between histological and functional evaluations. Mizutari et al. (2013) reported significant recovery of wave I amplitudes in ABRs after the application of a γ-secretase inhibitor [51], suggesting that hearing recovery by a γ-secretase inhibitor may be induced by the regeneration or preservation of ribbon synapses, not by OHC regeneration. Based on these experimental findings, we presume that a clinical trial of PIPE-505 (NCT04462198) was attempted to evaluate the efficacy of ribbon synapse regeneration by estimating speech-in-noise hearing impairment. Another possible explanation for the lack of significant recovery in DPOAEs in Tona et al. (2014) [52] is the insufficient maturity of the regenerated OHCs. Recent studies have suggested that newly generated HCs by ATOH1 forced expression lacked the morphological and electrophysiological features of mature HCs [53,54,55,56]. OHCs are very specifically differentiated cells, and their electrical motility plays a critical role in amplifying sound vibrations, called cochlear amplifiers. This indicates difficulty in regenerating fully differentiated OHCs, which may require specific cues.

In summary, these clinical trials demonstrated that intracochlear delivery of adenovirus vectors and intratympanic injections of γ-secretase inhibitors are safe and tolerable. However, meaningful hearing recovery by simple ATOH1 forced expression was not expected. The data on placebo controls using these intervention approaches are extremely valuable for designing future clinical trials using similar application routes, especially for setting the sample size.

### 4.3. Clinical Trials of Supporting Cell Reprogramming

Unique clinical trials (NCT03616223, 04120116, 4601909, 4629664, and 05086276) aiming for HC regeneration are underway, in which the main machinery is not the direct conversion of SCs to HCs, but the gain of stem cell potential in SCs. A set of seven small-molecule compounds, namely valproic acid (VPA, an HDAC inhibitor), FSK (forskolin, an adenylyl cyclase activator), CHIR (aminopyramidine derivative, a GSK-3β inhibitor), 616452 (a TGF-β receptor inhibitor), Tranyl (a histone demethylation inhibitor), DZNep (adenosine analog, an EZH2 inhibitor), and TTNPB (retinoic acid analog, a retinoic acid pathway activator), could replace all four Yamanaka transcription factors to successfully reprogram mouse somatic cells into pluripotent stem cells [57]. Among these seven small-molecule compounds, two compounds, VPA and CHIR, have been utilized for reprogramming mouse SCs [58]. Clinical trials were conducted based on this experimental study.

McLean et al. (2017) demonstrated the efficacy of the combination of VPA and CHIR (FX-322) in enhancing the SC potential for sphere formation and HC generation using cell cultures and for SC conversion to HCs in explants of neonatal mouse cochleae [58]. However, there are no data showing functional recovery in adult animals following the local applications of VPA and CHIR. The mechanisms underlying the effects of VPA and CHIR on HC regeneration are summarized by Samarajeewa et al. (2019) [59]. A GSK-3β inhibitor induces the stabilization of β-catenin in the cytoplasm, resulting in the activation of the canonical Wnt pathway, which plays an important role in the development of the cochlear sensory epithelium, similar to the Notch signaling pathway [59]. HDAC inhibition by VPA aims to improve chromatin accessibility in SCs because epigenetic decommissioning of HC-specific enhancers in mature SCs contributes to the loss of transdifferentiation capacity [60].

Clinical trials for the topical application of VPA and CHIR99021 (FX-322) are underway to investigate its potential for hearing restoration in patients with age-related hearing loss, noise-induced hearing loss, and sudden SNHL (Table 2). The outcome measures of these clinical trials are preferable. A Phase 1/2 safety trial (NCT03616223) demonstrated that a combination of VPA and CHIR99021 (FX-322) was safe and well tolerated following a single intratympanic injection with no serious adverse events [61]. Pharmacokinetics in animals and humans confirmed the intra-cochlear delivery of the agents after intratympanic injections [61]. Surprisingly, some patients exhibited clinically meaningful improvements (absolute word recognition improved by 18–42%) [61]. A Phase 2b clinical trial (NCT05086276) has been completed recently.

## 5. Future Directions toward Hair Cell Regeneration

Recent studies have revealed that forced ATOH1 expression alone is not sufficient to induce HC regeneration in mature mammalian cochleae [54,62]. These studies illustrate that the ability of SCs to transdifferentiate into HCs in response to ATOH1 forced expression is rapidly diminished by aging, especially after postnatal day 5 in mice [47,51]. This may be a reason for the unfavorable outcomes of auditory function assessments in clinical trials using simple ATOH1 forced expression. To promote the potential of SCs for transdifferentiation into HCs, the exploration of additional transcription factors to ATOH1 is necessary, and more precise mechanisms of SC-to-HC conversion are to be unveiled.

### 5.1. Co-Expression of Additional Transcription Factors to ATOH1

The age-dependent decline in the conversion ability of ATOH1 has led to the search for additional transcription factors to convert older cochlear SCs into HCs [63]. In general, the induction of a specific master gene for a desired cell type is crucial for fate conversion. In the case of HC regeneration, ATOH1 is known as the master gene for HCs. In the efficient direct conversion of other systems, the combination of multiple transcription factors is used [64]. Among the combination factors, pioneer factors play a key role in efficient conversion. Pioneer factors are a type of transcription factor that can bind and open closed chromatin to enable the binding of other canonical transcription factors [65,66]. Therefore, exploring pioneer factors specific to HC induction is key for successful SC-to-HC conversion in adult mammals.

Yamashita et al. (2018) performed RNA sequencing of newly converted HCs and mature OHCs after a conditional overexpression of ATOH1 in specific types of SCs, pillar and Dieters cells [67], which can be a source of newly converted HCs in the OHC region. As a result, they identified several transcription factors, including *Isl1*, *Ikzf2*, *POU4F3*, *LIN28B*, *Sall1/3*, and *Aff3* as lower-expressed genes in converted HCs than in mature OHCs [67]. These transcription factors are candidates for pioneer factors. Yamashita et al. (2018) showed the promotion of ATOH1-dediated SC-to-HC conversion by co-transfection *Isl1* but did not demonstrate the induction of converted HCs expressing mature OHC markers, including prestin [67]. Sun et al. (2021) demonstrated that the dual forced expression of ATOH1 and Ikzf2 promotes SC conversion to HCs expressing prestin in adult mammalian cochleae; however, the induced OHC-like cells remained immature [68].

ATOH1’s function is highly context dependent. During inner ear development, ATOH1 can only promote HC differentiation at specific developmental stages [69,70]. GFI1 and POU4F3 are two hair cell-specific transcription factors expressed downstream of ATOH1 during development and have been implicated in HC survival and function [71,72,73,74,75]. POU4F3 plays a major role in the maturation and survival of HCs, and GFI1 regulates HC differentiation by acting as a co-activator of ATOH1 and repressing non-HC genes [76]. Hence, GFI1 and POU4F3 have attracted attention as pioneer factors in ATOH1-dedicated HC regeneration. Lee et al. (2020) demonstrated that combinatorial ATOH1 and GFI1 induction enhanced SC-to-HC conversion in the adult mouse cochlea in vivo [77]. However, functional recovery was not observed. Chen et al. (2021) reported the generation of mature and functional HCs by co-expression of GFI1, POU4F3, and ATOH1 in postnatal mouse cochlea [78], suggesting that co-transfection of GFI1 and POU4F3 contributes to the maturation of converted HCs. More recently, Iyer et al. (2022) also demonstrated a higher efficiency of co-transfection of GFI1, POU4F3, and ATOH1 for SC-to-HC conversion than that of co-transfection of GFI1and ATOH1 or transfection of ATOH1 alone [79]. Walters et al. (2017) showed that a combination of ATOH1, Gata3, and POU4F3 improved the efficiency of HC regeneration in adult mice [56]. Additionally, this study demonstrated that POU4F3 upregulation is critical for SC-to-HC conversion [56]. A recent study also demonstrated that POU4F3 acts as a pioneer factor in enhancing the accessibility of HC loci in SCs [80]. These findings suggest the potential of POU4F3 as a pioneer factor in the induction of SC-to-HC conversion in adult mammalian cochleae. However, the efficiency of co-transfection with GFI1, POU4F3, and ATOH1 was also decreased by aging [79]. One of the major causes of this is a decrease in the accessibility of HC loci in mature SCs [79].

Successful direct conversion using co-transfection of pioneer factors in other systems has been reported [63,81]. Together with the regulation of chromatin accessibility, these strategies will become mainstream for inducing SC-to-HC conversion in adult mammalian cochleae.

### 5.2. Intermediate Progenitor State during Direct Conversion

Many mammalian organs lack stem cell pools similar to mature cochleae but sometimes exhibit regenerative capacity. Differentiated cells in organs accomplish the regeneration of not only their own cells but also different cell types via direct conversion [81,82]. Recent progress in studies focusing on the molecular mechanisms for direct conversion indicated that truly direct conversion, which bypasses developmental states during lineage conversion, is improbable and insufficient and may require the transient acquisition of the intermediate progenitor state for efficient conversion [81]. Leaman et al. (2022) proposed that the existence of intermediate cell states during reprogramming is a key element for successful fate conversion [81]. Therefore, the characterization of the intermediate progenitor state during fate conversion is important. However, little is known about the existence and characteristics of intermediate progenitor populations during SC-to-HC conversion. Understanding the characteristics of the intermediate progenitor state will help develop more efficient methods for the fate conversion of SCs to HCs.

Doetzlhofer and colleagues published a series of publications stating the roles of LIN28B and follistatin (FST) in the ability of mouse SCs to HC regeneration [83,84,85]. A recent study demonstrated that LIN28B and FST play key roles in the strict regulation of transforming growth factor beta (TGFβ) signaling, which is required for the induction of SC reprogramming [85]. LIN28B activates TGFβ signaling, and FST suppresses excessive activation of TGFβ signaling [85]. They used Hmga2, which is expressed in prosensory cells in the murine cochlea [86], as a marker for the intermediate progenitor state in neonatal mouse cochleae [84,85]. Co-activation of LIN28B and FST enables stage postnatal day 13 SCs to form HCs in organoid cultures and enhances the capability of SCs for HC regeneration in the neonatal cochlea [85]. This is a novel approach that focuses on the capability of SCs to convert to an intermediate progenitor state. However, the efficacy of the co-activation of LIN28B and FST in mature mammalian cochleae for HC regeneration is still unclear, and the molecular characteristics of the intermediate progenitor state have not been fully elucidated.

In the avian cochlea, HC regeneration spontaneously occurs after HC damage through SC-to-HC conversion and mitotic proliferation of SCs [87], similar to that in the neonatal mouse cochlea [88]. Matsunaga et al. established an explant culture model of HC regeneration via SC-to-HC conversion of chick cochleae [89] and explored molecular mechanisms of SC-to-HC conversion using single-cell RNA sequencing [90]. During SC-to-HC conversion in chick cochleae, SCs once de-differentiated into the intermediate progenitor state, which is characterized by the expression of *EDNRB2*, followed by the acquisition of HC identity [90]. In addition, suppression of SCs for the transition to the intermediate progenitor state by the inhibition of TGFβ signaling resulted in a decrease of regenerated HCs [90]. Elucidation of mechanisms for the transition from SCs to the intermediate progenitor state will contribute to the progress in strategies for inducing HC regeneration in mature mammalian cochleae.

## 6. Perspective for the Development of Future Pharmacotherapy for Hair Cell Regeneration

In the last decade, research and development of pharmacotherapy for SNHL have rapidly progressed. The major mechanism for HC protection against cisplatin is the reduction of cellular stress, which is a common cause of cochlear degeneration due to various causes. Therefore, we can expect to explore therapeutics that are commonly used for cochlear protection based on clinical trials of cisplatin-induced hearing loss. The progression of cochlear cell damage, particularly of the SCs in the sensory epithelium, can diminish the opportunities for HC regeneration. In fact, cisplatin also damages SCs [91]. Hence, cochlear protection is also important for the maintenance of sources for regenerated HCs.

Recently, clinical trials for stable SNHL have rapidly emerged [11,24]. The main target of these clinical trials is HC regeneration. In clinical trials targeting HC regeneration, local applications, including intratympanic and intracochlear applications, are predominantly used for drug administration [11,24]. The information on the safety and tolerability of these application routes, especially intracochlear application, is valuable. This approach will be utilized in future gene therapy.

Conversely, significant hearing recovery has not been reported in clinical trials for HC regeneration, although McLean et al. (2021) reported some beneficial effects [61]. In addition, no experimental studies have reported sufficient HC regeneration and hearing restoration in adult mammals. This indicates that further experimental studies are required to develop efficient strategies for inducing HC regeneration in the adult mammalian cochlea. For this purpose, precise analyses of the process for naturally occurring HC regeneration in the avian and fish auditory epithelia may be useful. Recent studies using single-cell RNA sequencing provide gene expression profiles of chick and zebrafish auditory epithelial cells in quiescent, degenerative, and regenerative status [90,92,93,94,95,96,97]. In contrast to mammalian cochleae, mammalian vestibular sensory organs retain the capacity for HC regeneration. Recent publications of mouse utricles demonstrated gene expression profiles during HC regeneration at a single-cell level [98,99]. Based on these data, practical strategies for inducing HC regeneration will be developed for mature mammalian cochleae.

## Figures and Tables

**Table 1 pharmaceutics-15-00777-t001:** Clinical trials of protection against cisplatin-induced hearing loss.

Study ID	Drug Name	Molecule	Mechanism	Route	Patient Population	Phase
NCT00652132	Pedmark	Sodium Thiosulfate	Chelator, Antioxidant	Intra-venous	1 month–18 years old Newly diagnosed with hepatoblastoma	Ph3, completed No results posted FDA approved (September 2022)
NCT00716976					1–18 years old Newly diagnosed with germ cell tumor, hepatoblastoma, medulloblastoma, neuroblastoma, osteosarcoma, or other malignancy	Ph3, completed Results posted
NCT04262336	DB-020	Sodium Thiosulfate	Chelator, Antioxidant	Intra-tympanic	≧18 years old	Ph1b, active, not recruiting
NCT01451853	SPI-1005	Ebselen	Antioxidant Glutathione-mimic	Oral	18–70 years old Diagnosed with hematologic malignancies and adult solid tumors	Ph2, unknown status
NCT05628233	SENS-401	R-azasetron	Calcineurin antagonist	Oral	≧18 years old	Ph2, not yet recruiting
NCT04915183	Lipitor	Atorvastatin	HMG CoA reductase inhibitor	Oral	≧18 years old Diagnosed with squamous cell carcinoma of the head and neck	Ph3, not yet recruiting

HMG CoA: hydroxymethylglutaryl-CoA, IV: intravenous application, IT: intratympanic application, Ph: phase, SNHL: sensorineural hearing loss.

**Table 2 pharmaceutics-15-00777-t002:** Clinical trials of regenerative medicine for sensorineural hearing loss.

Study ID	Drug Name	Molecule	Mechanism	Route	Patient Population	Phase
NCT02132130	CGF166	Adenovirus 5-ATOH1	HC regeneration	Intra-cochlear	Severe-to-profound SNHL 18–65 years old	Ph1/2a, completed Results posted
2016-004544-10 (EudraCT)	LY-3056480	γ-secretase inhibitor	HC regeneration	Intra-tympanic	Mild to moderate SNHL 18–84 years old	Ph1/2a, completed Results posted
NCT05061758					Stable SNHL 18–65 years old	Ph2b, Withdrawn
NCT04462198	PIPE-505	γ-secretase inhibitor	HC regeneration Synapse regeneration	Intra-tympanic	Bilateral SNHL 18–75 years old	Ph1/2a, completed No results posted
NCT04129775	OTO-413	BDNF	Synapse regeneration	Intra-tympanic	Normal hearing or moderately severe hearing impairment 21–64 years old	Ph1/2a, completed No results posted
NCT03616223	FX-322 (CHIR99021 + Valproic acid)	GSK3 inhibitor + HDAC inhibitor	HC regeneration	Intra-tympanic	Stable SNHL 18–65 years old	Ph1/2a, completed Results posted
NCT04120116					Stable SNHL 18–65 years old	Ph2a, completed No results posted
NCT05086276					Acquired, adult-onset, SNHL associated with noise-induced SNHL or idiopathic sudden SNHL 18–65 years old	Ph2b, completed, No results posted
NCT04601909					Age-related SNHL 66–85 years old	Ph1b, completed No results posted
NCT04629664					Acquired (non-genetic) severe SNHL 18–65 years old	Ph1b, completed No results posted

BDNF: brain-derived neurotrophic factor, GSK3: glycogen synthase kinase 3, HDAC: histone deacetylase, Ph: phase, SNHL: sensorineural hearing loss.

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
