# Peer review of "Future Pharmacotherapy for Sensorineural Hearing Loss by Protection and Regeneration of Auditory Hair Cells"

_pharmaceutics, 2023, doi:10.3390/pharmaceutics15030777_

Round 1

Reviewer 1 Report

This is a review article of clinical trial advances for treating hearing loss, specifically hair cell regeneration. This review covers systemic or transtympanic or injected molecules to protect or restore hearing. It does not cover gene replacement therapy, which is very promising. It may be worth stating in the article that the scope of the review is to not cover gene replacement therapy.

On page 3, I would recommend make a statement about the effect size of sodium thiosulfate in cisplatin trials (hearing loss 39% vs. 68% in the SIOPEL 6 trial and 44% vs. 58% in the COG ACCL0431 trial).

The authors conclude on page 6 that “data on placebo controls using these intervention approach are extremely valuable for future trails”.  Yes, controlled trial is need but only if there is some efficacy seen with the drug alone.  ATOH1 and gamma-secretase inhibitors failed to show any benefit, so a control trial is not needed.

Overall, this article touches on animal studies and some preliminary human clinical trials. While there have been limited advances in the clinic, the article sufficiently covers the active trials.

Typo:

Page 1: (HCs) form four rows

Page 6: Atoh1-dedicated

Author Response

Reviewer 1

This is a review article of clinical trial advances for treating hearing loss, specifically hair cell regeneration. This review covers systemic or transtympanic or injected molecules to protect or restore hearing. It does not cover gene replacement therapy, which is very promising. It may be worth stating in the article that the scope of the review is to not cover gene replacement therapy.

We appreciate the reviewer for valuable comments and suggestions. According to this suggestion, we have stated the scope of gene therapy in Section 2 (line 80-85, page 2).

On page 3, I would recommend make a statement about the effect size of sodium thiosulfate in cisplatin trials (hearing loss 39% vs. 68% in the SIOPEL 6 trial and 44% vs. 58% in the COG ACCL0431 trial).

We have stated these results of phase 3 clinical trials for sodium thiosulfate in line 130-135, page 4.

The authors conclude on page 6 that “data on placebo controls using these intervention approach are extremely valuable for future trails”.  Yes, controlled trial is need but only if there is some efficacy seen with the drug alone.  ATOH1 and gamma-secretase inhibitors failed to show any benefit, so a control trial is not needed.

We consider that outcomes of placebo controls in these clinical trials using intra-cochlear application or intratympanic injections are valuable, because of two reasons. One is future contribution to setting of the sample size for future clinical trials using similar methods for drug application. Another is to show the safety of these drug delivery methods, especially for future gene replacement therapy. We have revised as ‘The data on placebo controls using these intervention approaches are extremely valuable for design future clinical trials using similar application routes, especially for setting the sample size.’ (Line 263-265) and ‘This approach will be utilized in future gene therapy’. Line 414-415).

Overall, this article touches on animal studies and some preliminary human clinical trials. While there have been limited advances in the clinic, the article sufficiently covers the active trials.

Thank you for your positive comment.

Typo:

Page 1: (HCs) form four rows

Page 6: Atoh1-dedicated

We have corrected these errors.

Reviewer 2 Report

Summarize the main findings of the study:

In the present manuscript entitled “Future pharmacotherapy for sensorineural hearing loss by hair cell regeneration”, the authors aim to provide a comprehensive overview of clinical trials of pharmacotherapy for sensorineural hearing loss. The authors summarized recent clinical trials for the prevention of sensorineural hearing loss and regenerative medicines, as well as some experimental studies based on hair cell regeneration mechanisms, including SC-to-HC conversion and intermediate progenitor state. Overall, this review is of interest and generally comprehensive. However, there are still some issues that need to be addressed.

Major suggestions:

1.     It is a little confusing that the title and the abstract of the manuscript did not mention trials for the prevention of SNHL, which might also be one of the important parts of the text.

2.     As the review focuses on clinical trials, the summary of clinical trial-related content is less, insufficiently in-depth, and incomprehensive. For example, preclinical data on effectiveness and safety; the age and the cause and degree of the subject’s hearing loss, etc.

3.     The structure of the article seems not very clear. Points 5, 6, and 9 seem the subtitle of 4. Points 7 and 9 seem to belong to experimental studies based on hair cell regeneration mechanisms rather than clinical trials.

4.     It would be better to summarize the etiologies of sensorineural hearing loss earlier in the article, such as genetics, noise, ototoxic drugs, and aging, for readers to understand the manuscript better.

5.     If the results of clinical trials have been posted, it is recommended to add a summary of the result in Table 1 and Table 2.

Minor tips:

1.     The definition of SNHL needs to be described in section 1.

2.     In lines 34 to 39, the significance of pharmacotherapy studies for sensorineural hearing loss should not be limited to economics. The drawbacks of existing treatment options or the benefits of pharmacotherapy for patients should be discussed.

3.     What is the definition of “stable SNHL” in line 33? Do you mean the opposite of acute SNHL?

4.     Some clinical trials for regenerative medicines did not mention what types of SNHL they are applied to.

Author Response

Reviewer 2

In the present manuscript entitled “Future pharmacotherapy for sensorineural hearing loss by hair cell regeneration”, the authors aim to provide a comprehensive overview of clinical trials of pharmacotherapy for sensorineural hearing loss. The authors summarized recent clinical trials for the prevention of sensorineural hearing loss and regenerative medicines, as well as some experimental studies based on hair cell regeneration mechanisms, including SC-to-HC conversion and intermediate progenitor state. Overall, this review is of interest and generally comprehensive. However, there are still some issues that need to be addressed.

Thank you for your positive comments.

Major suggestions:

  1. It is a little confusing that the title and the abstract of the manuscript did not mention trials for the prevention of SNHL, which might also be one of the important parts of the text.

According to the comment, we have revised a title as ‘Future pharmacotherapy for sensorineural hearing loss by protection and regeneration of auditory hair cells’.

  1. As the review focuses on clinical trials, the summary of clinical trial-related content is less, insufficiently in-depth, and incomprehensive. For example, preclinical data on effectiveness and safety; the age and the cause and degree of the subject’s hearing loss, etc.

According to this comment, we have added ‘Patient population’ in tables. As for the trials that results were obtained, we have stated the results in the main text.

  1. The structure of the article seems not very clear. Points 5, 6, and 9 seem the subtitle of 4. Points 7 and 9 seem to belong to experimental studies based on hair cell regeneration mechanisms rather than clinical trials.

According to this suggestion, sections 5, 6, and 9 have arranged as subsection 4.1, 4.2, and 4.3. Sections 7 and 8 are combined as section 5 (Future directions toward hair cell regeneration) and arranged as subsections 5.1 and 5.2. We have added the description of the intermediate progenitor state during HC regeneration in chick cochleae in subsection 5.2 (line 388-398).

  1. It would be better to summarize the etiologies of sensorineural hearing loss earlier in the article, such as genetics, noise, ototoxic drugs, and aging, for readers to understand the manuscript better.

According to this suggestion, we have stated the etiologies of sensorineural hearing loss in Section 1 (line 26-29).

  1. If the results of clinical trials have been posted, it is recommended to add a summary of the result in Table 1 and Table 2.

The number of clinical trials that are posted results is limited. We have stated these results in the main text (line 130-135, 181-183, and 297-299).

Minor tips:

  1. The definition of SNHL needs to be described in section 1.

We have stated it in section1 (line 26-28).

  1. In lines 34 to 39, the significance of pharmacotherapy studies for sensorineural hearing loss should not be limited to economics. The drawbacks of existing treatment options or the benefits of pharmacotherapy for patients should be discussed.

According to this suggestion, we have stated drawbacks of hearing devices in the second paragraph of Section 1 (line 42-45).

  1. What is the definition of “stable SNHL” in line 33? Do you mean the opposite of acute SNHL?

We have stated the definition of stable SNHL in the in the second paragraph of Section 1 (line 38-41).

  1. Some clinical trials for regenerative medicines did not mention what types of SNHL they are applied to.

In the revised version, we have stated ‘Patient population’ in Table 2.

Reviewer 3 Report

Hair cell loss-caused sensorineural hearing is irreversible due to the lack of regeneration in mammalian cochlea; thus, how to protect sensory hair cells from damages and how to promote hair cell regeneration are two important yet unsolved questions in the inner ear field. The authors of the manuscript “Future pharmacotherapy for sensorineural hearing loss by hair 2 cell regeneration” summarized current clinical trials of hearing loss prevention and hair cell regeneration. They also discussed the molecular mechanisms underneath these clinical trials and speculated the possible reasons for the unsatisfactory results. The manuscript was written well, and the major problem of the field was covered. The discussion was organized in a way easy to understand and should be of interest for general audience too. Below are a few issues to make this manuscript better.

                At the end of section 2, the authors stated that hair cell loss is considered as a major cause of sensorineural hearing loss and hair cells are the major targets for the treatment. Though the authors reasoned that mutations in many deafness genes contribute to SNHL, the sensitivity of hair cells to environmental factors such as noise and drugs is also another reason people fucus on hair cell protection and regeneration. It should be discussed.

                In section 3, clinical trials for drugs against cisplatin ototoxicity were discussed but research works on hair cell protection against other forms of hearing loss also worth a mention, especially there are many researchers focusing on aminoglycoside-induced hearing loss and hair cell protection, as well as noise-induced hearing loss.

                In section 5 for ATOH1 gene therapy, the authors discussed the different outcomes between animal works and the clinical trail, and the discrepancy among different groups. The authors reasoned that whether responsible supporting cells remain in the cochleae is one potential factor contributing to the failure of clinical trial. Actually, there are multiple papers, including papers cited in this manuscript, showing that regenerated hair cells by forced expression of Atoh1 are immature and non-functional; and it is believed that the failure of regenerating mature hair cells is the reason behind the lack of functional recovery. This opinion should also be discussed. Actually, persist expression of Atoh1 in hair cells leads to hair cell death (Liu et al., 2012); therefore, how to force the expression of Atoh1 in supporting cells for cell fate conversion and then shut Atoh1 down for hair cell maturation might be one of the topics for future research/trials. It would be better to include it in the perspectives.

                For section 7, it would be better to name the section as “Co-expression of additional transcription factors” since not all the factors discussed in this section are pioneer factors. By definition, pioneer factors are transcription factors that can bind to closed chromatin and then activate elements for transcription regulation. There is little or no evidence for the pioneer factor activity for Ikzf2 or Gfi1, but the pioneer factor activity of Pou4f3 and Isl1 has been shown.

                I personally think it would be better to merge section 8 and 9. Though different approaches were discussed, genetic manipulation in section 8 but small chemicals in section 9, both sections were focused on how to roll supporting cells back to an undifferentiated progenitor state.

For the last section, the authors stated that revisiting the avian model of would benefit current research work on hair cell regeneration. Besides the avian model, other models should also be mentioned, such as the regeneration of sensory hair cells in fish, and natural/forced regeneration in mouse/human utricles.

Author Response

Reviewer 3

Hair cell loss-caused sensorineural hearing is irreversible due to the lack of regeneration in mammalian cochlea; thus, how to protect sensory hair cells from damages and how to promote hair cell regeneration are two important yet unsolved questions in the inner ear field. The authors of the manuscript “Future pharmacotherapy for sensorineural hearing loss by hair cell regeneration” summarized current clinical trials of hearing loss prevention and hair cell regeneration. They also discussed the molecular mechanisms underneath these clinical trials and speculated the possible reasons for the unsatisfactory results. The manuscript was written well, and the major problem of the field was covered. The discussion was organized in a way easy to understand and should be of interest for general audience too. Below are a few issues to make this manuscript better.

Thank you for your positive comments and understanding our aim of this review.

                At the end of section 2, the authors stated that hair cell loss is considered as a major cause of sensorineural hearing loss and hair cells are the major targets for the treatment. Though the authors reasoned that mutations in many deafness genes contribute to SNHL, the sensitivity of hair cells to environmental factors such as noise and drugs is also another reason people fucus on hair cell protection and regeneration. It should be discussed.

We have described this issue at the end of Section 2 (line 80-85).

                In section 3, clinical trials for drugs against cisplatin ototoxicity were discussed but research works on hair cell protection against other forms of hearing loss also worth a mention, especially there are many researchers focusing on aminoglycoside-induced hearing loss and hair cell protection, as well as noise-induced hearing loss.

Exactly. We should describe our previous efforts. According to this comment, we have added the second paragraph of Section 3 describing the importance of previous studies on aminoglycoside- and noise-induced damage to hair cells (line 91-102).

                In section 5 for ATOH1 gene therapy, the authors discussed the different outcomes between animal works and the clinical trail, and the discrepancy among different groups. The authors reasoned that whether responsible supporting cells remain in the cochleae is one potential factor contributing to the failure of clinical trial. Actually, there are multiple papers, including papers cited in this manuscript, showing that regenerated hair cells by forced expression of Atoh1 are immature and non-functional; and it is believed that the failure of regenerating mature hair cells is the reason behind the lack of functional recovery. This opinion should also be discussed. Actually, persist expression of Atoh1 in hair cells leads to hair cell death (Liu et al., 2012); therefore, how to force the expression of Atoh1 in supporting cells for cell fate conversion and then shut Atoh1 down for hair cell maturation might be one of the topics for future research/trials. It would be better to include it in the perspectives.

We appreciate the reviewer for this valuable comment. According to this comment, we have stated the problem of persisted expression of Atoh1 in line 206-208.

                For section 7, it would be better to name the section as “Co-expression of additional transcription factors” since not all the factors discussed in this section are pioneer factors. By definition, pioneer factors are transcription factors that can bind to closed chromatin and then activate elements for transcription regulation. There is little or no evidence for the pioneer factor activity for Ikzf2 or Gfi1, but the pioneer factor activity of Pou4f3 and Isl1 has been shown.

We appreciate the reviewer for this valuable comment. The section title was revised as “Co-expression of additional transcription factors to ATOH1”.

                I personally think it would be better to merge section 8 and 9. Though different approaches were discussed, genetic manipulation in section 8 but small chemicals in section 9, both sections were focused on how to roll supporting cells back to an undifferentiated progenitor state.

We agree with this suggestion. However, Reviewer 2 asked to combine sections 5, 6, and 9, because these sections describe clinical trials. On the other hand, sections 7 and 8 describe experimental findings. In consequence, we merged sections 5, 6, and 9 and arranged as subsections from the point of view of recent clinical trials. Sections 7 and 8 are combined as section 5 in the revised version entitled ‘Future directions toward hair cell regeneration’ and arranged as subsections.

For the last section, the authors stated that revisiting the avian model of would benefit current research work on hair cell regeneration. Besides the avian model, other models should also be mentioned, such as the regeneration of sensory hair cells in fish, and natural/forced regeneration in mouse/human utricles.

We agree with this suggestion. We have stated this issue at the end of the last section (line 421-429).